# Children’s Perspectives on Using Serious Games as a Complement to Promoting Their Social–Emotional Skills

**DOI:** 10.3390/ijerph19159613

**Published:** 2022-08-04

**Authors:** Ana Xavier, Paula Vagos, Lara Palmeira, Paulo Menezes, Bruno Patrão, Sónia Pereira, Vanessa Rocha, Sofia Mendes, Marta Tavares

**Affiliations:** 1Instituto de Desenvolvimento Humano Portucalense, Universidade Portucalense Infante D. Henrique, 4200-072 Porto, Portugal; 2Faculty of Psychology and Educational Sciences, Center for Research in Neuropsychology and Cognitive and Behavioural Intervention (CINEICC), University of Coimbra, 3000-115 Coimbra, Portugal; 3Serviço de Psicologia Clínica (SPC), Centro Hospitalar e Universitário de Coimbra (CHUC), 3004-561 Coimbra, Portugal; 4Institute of Systems and Robotics (ISR), University of Coimbra, 3030-194 Coimbra, Portugal; 5Centro de Investigação em Psicologia para o Desenvolvimento, Instituto de Psicologia e Ciências da Educação, Universidade Lusíada, 4100-346 Porto, Portugal; 6Agrupamento de Escolas de Valadares, 4405-594 Valadares, Portugal

**Keywords:** social–emotional learning, serious games, children, quantitative and qualitative appraisal of intervention

## Abstract

The use of serious games may be an appealing and complementary way to motivate curriculum-based social and emotional learning (SEL); still, investigation into this potential usefulness is scarce. This study aims to address the usefulness of serious games within the program ‘Me and Us of Emotions’. Specifically, we analyzed the differences in children’s satisfaction in sessions that did or did not use serious games as a complement to the intervention, explored the contribution of using serious games to the global satisfaction with the program, and explored children’s qualitative feedback regarding the sessions. The participants were 232 children (122 boys and 110 girls) aged between 8 and 12 years old (*M* = 9.09, *SD* = 0.80). The measures were based on the subjective appraisals of the sessions made by the participating children, including quantitative and qualitative assessments of the degree of satisfaction of the participants. The results showed that there were similar levels of satisfaction with the sessions that did or did not use serious games as a complement to the program. However, only satisfaction with the sessions that used serious games (and not satisfaction with the sessions that did not use them) contributed significantly to explaining both the enjoyment of the activities and the interest in the subjects. Satisfaction with serious games was significantly and positively associated with fun, easiness, ability to understand the session, and ability to cope with emotions. Qualitative analysis showed three main themes, namely: positive aspects, negative aspects, and opportunities for improvement of the program. Overall, these results indicate that children’s satisfaction with the ‘Me and Us of Emotions’ program is related to serious games, suggesting the relevance of using this complementary tool more often when intervening with younger generations.

## 1. Introduction

The empirical literature has consistently established the importance of universal and preventive approaches to promoting social and emotional skills in children within school contexts [1,2,3,4]. Promoting these social and emotional skills is based on the Social and Emotional Learning (SEL) framework. SEL is operationalized as “the process through which children and adults develop the skills, attitudes, and values necessary to acquire social and emotional competence” [5] (p. 2). Meta-analytic findings show that evidence-based SEL programs applied to elementary, middle, and high schools have proven to be beneficial in increasing well-being, prosocial behaviors, and academic performance and in reducing externalizing and internalizing of problems [1,4,6,7]. According to the CASEL consortia [8,9], social–emotional skills include five key competencies, namely, self-awareness, self-management, social awareness, relationship management, and responsible decision making [2]. These competencies can be taught, modeled, and learned through instruction, practice, and feedback [10].

SEL also encourages prosociality and compassionate motives [11,12]. According to compassion-focused therapy [13,14], human beings are intrinsically motivated to defend themselves from threats, seek resources, give care, and cooperate. These motivations are linked to different but related affect regulation systems, namely, threat, drive, and contentment/affiliation. This latter system is associated with feelings of calm, peacefulness, security, and safety [14] and with bonding behaviors and prosocial actions toward others. Compassion is rooted in this caregiving mentality and is a prosocial motivation that has positive effects on educational climate and academic performance [15,16,17]. Particularly, empathy, compassion, and cooperation are linked to psychological well-being in adolescents [18,19] and adults [20]. Additionally, compassion-based interventions have benefits for both intra- and interpersonal relationships and seem to promote not only compassion but also self-regulation abilities in diverse populations and settings [21], including in the school context [15,16,22]. Despite its beneficial results, programs that complement SEL with compassionate components are still scarce in young populations or educational settings.

The majority of SEL programs have been traditionally taught within the curriculum [3,4]. In recent years, serious games have been incorporated into those programs [4]. Serious games are video games used with an educational purpose (e.g., training, knowledge acquisition, skills development; [23]). In education, serious games have more generally been used for academic purposes and have proven to be effective in enhancing cognitive abilities and positive attitudes toward learning [24,25]. Using serious games for social–emotional learning is rarely done and mainly targets clinical samples (such as training empathy for children with autism spectrum disorder; [26]). In the general population, mixed results were found in the few studies that used serious games to promote social skills, as part of SEL [23,27]. Although some studies report the independent positive effects of serious games in promoting social skills, others indicate that serious games are beneficial when paired with in-person guided discussion [27]. Altogether, these studies stressed that continuing research on the usefulness of serious games as a complement to SEL programs is needed.

One way to measure the usefulness of serious games relies on assessing participant engagement, for example, the extent to which participants are sufficiently attracted to and involved in the program [28], which can be measured by the degree of satisfaction of the participants. Assessing the quality of program implementation based on the indicators of feasibility and responsiveness is considered good practice [29,30] and an essential condition of effective social and emotional learning programs [31,32]) and ensures that quality has also been associated with the interventions’ outcomes [33]. When nuclear elements of fidelity were separately considered, participant responsiveness (i.e., involvement and engagement of children/adolescents) was significantly associated with SEL outcomes post-test [34]. Additionally, qualitative evaluations of program implementations can also inform about their quality and variability [33]. A previous study combining quantitative and qualitative research indicates that participant-type factors associated with implementation, such as participants’ engagement, attitudes, and motivation, as well as program factors, such as lengthening the program, are key to the implementation quality of an intervention [35]. Thus, qualitative feedback from participants can inform about the acceptability of the programs and provide information on what was learned and what were the strategies that participants perceived as more helpful [36].

Despite the relevance of knowing about participants’ engagement with the interventions to better interpret the outcomes and continuously improve the interventions, it is a rare practice in the SEL literature [31,33]. It becomes even more relevant when the SEL programs are innovative, designed based on the specificities of the participants’ development and make use of serious games as a complement to intervention delivery. So, the present study aimed to investigate the participants’ perceived responsiveness and quality of an SEL-based intervention program designed for children, the ‘Me and Us of Emotions’ program, which uses serious games as a complement to face-to-face and interactive intervention sessions. Specifically, we aimed to compare children´s satisfaction with intervention sessions that did or did not use serious games as a complement to the intervention and explored two sessions that used serious games in relation to fun with, easiness, and utility. Furthermore, we intended to explore how using serious games in the sessions contributes to being satisfied with the whole program. Finally, the current study also aimed to explore children’s qualitative feedback regarding the sessions. We expect quantitative and qualitative data to converge in that using serious games provides higher satisfaction, given that previous works found that such activities are associated with engagement and motivation [27].

## 2. Method

### 2.1. Participants

Participants were 232 children aged 8–12 years old (*M* = 9.09, *SD* = 0.80) of whom 52.6% (*n* = 122) were boys and 47.4% (*n* = 110) were girls. There were gender differences for age, *t*(230) = 2.74, *p* = 0.007, Cohen’s d = 0.360, with boys being significantly older than girls (*M* = 9.23, *SD* = 0.77 vs. *M* = 8.95, *SD* = 0.81). Regarding school year, 44.8% (*n* = 104) were in the 3rd grade and 55.2% (*n* = 128) were in the 4th grade. Boys and girls were evenly distributed by school year, *X*^2^(1) = 0.952, *p* = 0.329.

### 2.2. Procedure

#### 2.2.1. Sample Recruitment

Ethical approvals were obtained by the host institutions, after which two public schools in the northern region of Portugal were contacted to establish partnerships to recruit participants and implement the intervention in the school context. Schools asked parents for informed written consent for data collection. Both parents and children gave their written consent to collect data. Children enrolled in the study were fully informed about the goals of the study and the aspects of confidentiality. They agreed to participate and fill out the instruments voluntarily in the classroom in the presence of the teacher and at least one member of the research team. When necessary, clarification regarding the self-report protocol was provided. To compose the program’s groups, students within a class were randomly assigned to either an experimental group (EG) or a control group (CG). In the present study, only the data from the EG was used; participants in this group received the ‘Me and the Us of Emotions’ program.

#### 2.2.2. The Me and the Us of Emotions Program

The Me and the Us of Emotions program [37] comprises 10 manualized and developmentally appropriate weekly group sessions with a duration of 60 min each, which are delivered in the classroom. Four psychologists with previous training in the program and one clinical psychology master’s student implemented the sessions for each class in the presence of the teacher. The program is preferably delivered in person but can also be delivered online, if necessary. It was designed as a universal program fostering social–emotional skills that are well-suited for an educational audience in school contexts. The 10 sessions include psychoeducation on the physiological, cognitive, and behavioral components of emotions and how they are linked, regarding basic emotions (e.g., joy, sadness, fear, anger) and secondary emotions (e.g., self-reassurance and compassion). Active actions via manualized activities (e.g., drawing, writing) were also used as a way for children to individually appropriate their gained knowledge of emotions. Additionally, compassion-focused approaches through experiential exercises (e.g., exercises in imagery) were focused on promoting self-reassuring and self-soothing abilities and prosocial and compassionate behaviors (e.g., compassionate touch exercise from [18]; safe-place meditation; and compassionate letter exercise adapted from CFT, [38]). The attainment of these goals was complemented by the use of serious games in five sessions (particularly, 2, 7, 8, 9, and 10). In addition, these games were made available online and could be accessed from home and in between the sessions. These serious games, which were specifically developed for the program, focused on specific emotions (e.g., joy, sadness, fear, anger) and included the characters of the program that the user must guide throughout the scenario to collect reward items (i.e., coins representing useful strategies to deal with the difficult emotion) and avoid obstacles (representing ineffective strategies to deal with the difficult emotion). Thus, serious games were included as a complementary tool to sessions with an educational purpose.

### 2.3. Measures

#### 2.3.1. Satisfaction with Each Session

Satisfaction with the program was assessed weekly after each session by each participating child. Children answered the question “How much did you like this session?” using a 5-point Likert response scale ranging from “I didn’t like it at all” to “I liked it a lot”. The total satisfaction scores by type of session were computed by summing answers provided to this item across sessions 2, 7, 8, 9, and 10 (i.e., satisfaction with the sessions that used serious games) and across sessions 1, 3, 4, 5, and 6 (i.e., satisfaction with the sessions that did not use serious games).

#### 2.3.2. Satisfaction with Serious Games

After sessions 9 and 10, both using serious games, participating children answered 4 questions: fun with serious games (“How often did you have fun playing this game?”); easiness (“How easy was it to understand how to play?”); usefulness in relation to understanding the session (“Did the game help you to understand the session?”); and usefulness in relation to understanding how to cope with emotion (“Did the game help you to understand how to deal with the emotion?”). All these questions were rated on a 5-point scale ranging from 1 “None” to 5 “Much”.

#### 2.3.3. Satisfaction with the Whole Program

This questionnaire included five items and was completed at the end of the program. The first item was “You think that the sessions were…” rated on a 5-point scale ranging from “Not good at all” to “Very good”. The other four items were “How would you rate the sessions?”; “Did you like the activities conducted during the session?”; “What is your interest in the subjects we talked about?”; “Did you like the music?”; and “Did you like the games?”, which were rated on a 5-point Likert scale ranging from “None” to “Very much”. In addition, four open-ended questions were included to be analyzed thematically (“Did you like the program? Explain why”; “What did you like most about the program?”; “What did you like least?”; and “What advice would you give to make the sessions go better?”).

### 2.4. Data Analysis

Statistical analyses were conducted using PASW Software (Predictive Analytics Software, version 27, SPSS, Chicago, IL, USA). Descriptive statistics were computed to explore the demographic variables and gender differences were tested using independent sample *t*-tests [39]. Additionally, a *t*-test was also used to compare the satisfaction between the sessions that did and did not use serious games. Pearson product–moment correlations were performed to explore the relationships between the variables in the study [39,40]. Multiple regression analyses using the standard method were conducted to explore the contribution of satisfaction with the sessions that did and did not use serious games as independent variables to the prediction of the variance of satisfaction with the whole program, particularly, enjoying the sessions (question 1), enjoying the activities (question 2), interest in the subjects (question 3), liking the music (question 4), and liking the games (question 5). In these quantitative analyses, a listwise approach was used to missing data due to data not being missing completely at random. Qualitative data analysis was explored with MAXQDA Analytics Pro 2020, which allowed to organize children’s feedback from the sessions into categories and sub-categories.

## 3. Results

### 3.1. Quantitative Data

#### 3.1.1. Differences in Satisfaction between Sessions with and without Serious Games

The results showed no differences in satisfaction between the sessions that did and did not use serious games (*n* = 59, *M* = 4.77, *SD* = 0.48 vs. *M* = 4.72, *SD* = 0.46, respectively), *t*(58) = 1.062, *p* = 0.292.

#### 3.1.2. Predictors of Satisfaction with the Whole Program

No significant effects of satisfaction with the sessions that did and did not use serious games were found in enjoying the sessions, *F*
_(2, 52)_ = 1.132, *p* = 0.330, liking the music, *F* _(2, 53)_ = 0.498, *p* = 0.611, and liking the games, *F* _(2, 55)_ = 0.970, *p* = 0.385.

Alternatively, satisfaction with the sessions that did and did not use serious games were significant predictors of the variance of enjoying the activities, *F* _(2, 54)_ = 19.416, *p* < 0.001, accounting for 39.7% of that enjoyment. Only satisfaction with the sessions that used serious games was a significant predictor. The same predictive model was significant for the interest in the subjects being covered in the session, *F* _(2, 54)_ = 15.325, *p* < 0.001, with satisfaction with the sessions that did and did not use serious games accounting for 33.8% of the variance of that interest. Again, only satisfaction with the sessions that used serious games emerged as a significant predictor (Table 1).

#### 3.1.3. Correlations between Satisfaction with, Fun with, Easiness, and Utility of the Serious Games in Sessions 9 and 10

Satisfaction with the use of serious games (in sessions 9 and 10) was significantly and positively associated with fun (*n* = 189, *r* = 0.64, *p* < 0.001), easiness (*n* = 194, *r* = 0.23, *p* = 0.002), ability to understand the session (*n* = 192, *r* = 0.56, *p* < 0.001), and ability to cope with emotions (*n* = 200, *r* = 0.61, *p* < 0.001).

### 3.2. Qualitative Data

Three main themes emerged from the analysis of the 221 participants who provided their feedback. They were (a) positive aspects, (b) negative aspects, and (c) opportunities for improvement to the program. In each major theme, participants’ responses were grouped into several sub-themes that best reflected participants’ verbalizations (Table 2).

Regarding the theme “positive aspects”, three sub-themes emerged, namely, learning emotion regulation strategies, experiencing positive emotions (through playing activities, using serious games, experiencing enjoyment, and experiencing a shared experience), and gaining knowledge about emotions (awareness of others’ emotions, body awareness, emotional expression). The theme “negative aspects” included a reduced opportunity for playing serious games and experiencing difficult emotions. Finally, the “opportunities for improvement” were the following: providing for less noise during the sessions and better behavior on the part of the participating children, addressing more emotions, using more serious games, and increasing the duration of the sessions.

## 4. Discussion

The value of universal social–emotional learning-based programs for children in school contexts is well-known [1,3,4,7] but the usefulness of serious games as complementary tools to those programs is still rarely studied [27]. In the ‘Me and the Us of Emotions’ program, serious games were used as a complement to some sessions addressing emotions (e.g., joy, anger) with a learning purpose (e.g., identifying emotions and the related useful behaviors to cope with emotions). The present study aimed to analyze whether using serious games as a complement to that program contributed to enhancing children’s satisfaction, as well as to explore the qualitative feedback from children who participated in the program.

The results showed that children reported similar levels of satisfaction with the sessions that did and did not use serious games as a complement to the program and that those diverse sessions did not differently impact on the enjoyment of the sessions, liking the music, and the liking games. The acceptability and the efficacy of traditional SEL programs have been previously proven [4,6] as has been preliminarily proven the efficacy of SEL programs using serious games [25]. Our findings seem to concur with the relevance of combining intervention strategies, as previously noted by [27], in as much as both traditional and serious games were satisfactory. Our qualitative results provided additional support for the positive effects of the interventions on children’s satisfaction. Overall, the participants considered the program as a way to learn emotion regulation strategies, experience positive emotions (through playing activities and serious games, and through experiencing enjoyment and a shared experience), and gain knowledge about emotions (awareness of others’ emotions, body awareness, emotional expression). Previous studies have found that similar participants’ feedback on their involvement and engagement was an indicator of the responsiveness to and acceptability of the program [35,36]. Additionally, a meta-analysis about serious games in education indicated that more positive attitudes toward learning occur when using serious games in comparison with traditional paper-based learning [25].

Alternatively, only satisfaction with the sessions that used serious games contributed to explaining both the enjoyment with the activities and the interest in the subjects; satisfaction with the sessions that did not use those games was not a significant contributor to that enjoyment and interest. So, although the children were similarly satisfied with all the sessions, using serious games helped them enjoy the activities more and be more interested in the sessions’ themes. Complementing sessions with serious games seemed to provide an appealing, interactive, and motivating environment, which aligns with previous literature that demonstrated that this complementary use of serious games could be a way of enhancing engagement and motivation in children [23].

In the current study, the two final sessions that used serious games were assessed by children regarding their fun, easiness, usefulness to understand the session, and usefulness in managing the emotion. The results suggest that children who were more satisfied with serious games tended to experience more fun and more easiness and found the games more useful for understanding the session and managing emotions. Although the limitations linked to the correlational nature of these findings, they provide some indication that when serious games are included in face-to-face social–emotional skills-based programs, they can facilitate children’s acquisition of knowledge and skills, which is consistent with the results from the few existent studies [23,27]. Also, the present findings are in accordance with a meta-analysis by [25] that showed that gaming easiness, perceived usefulness, and goal clarity are important factors for the effectiveness of serious games in knowledge acquisition focused on content and enjoyment.

The qualitative study also highlighted the importance of serious games as a complementary tool to the programs. The “negative aspects” highlighted by children were the reduced opportunity for playing serious games and experiencing difficult emotions. In addition, the “opportunities for improvement” included suggestions not only for ameliorating climate and behavioral manifestations during the sessions (e.g., less noise during sessions, behaving better in the classroom), but also for increasing the session duration, targeting more emotions, and using more serious games. It seems that both individual and program factors were highlighted by children both as positive and negative aspects and, consequently, as aspects that could be further improved. Similarly, participant factors have been emphasized in other studies, such as attitudes toward learning and motivation, as well as program factors, such as length [34,35].

Some limitations of the present study should be considered. The first limitation is related to the sample size. Although the current sample is considered large, not all participants consistently reported their satisfaction after each session (e.g., participant was absent in that session). In addition, the access to serious games was not equal for all children in the session due to time constraints. Although online access to serious games was possible through the ‘Me and the Us of Emotions’ program online platform, some children may not have had a sufficient internet connection at home. A further limitation was that the specific aspects concerning the fun, easiness, and usability of the serious games assessed in sessions 9 and 10 were not evaluated in the other sessions that used games; so, the findings may be specific to the games used in those sessions (and the cumulative effect of the intervention) and not to the use of serious games in general. Future studies should consider the same measurement methods over time. The control group was on a waiting list and not active, which limited the possible comparison of the complementary aspect of serious games between the groups. Future studies should also address the impact of serious games on participants’ emotional development. The current findings should be seen as a preliminary assessment of the potential usefulness of serious games as a complementary tool for social–emotional learning programs, and a larger study informed by these results is ongoing.

## 5. Conclusions

Both political and educational audiences have come to realize that a child’s social and emotional well-being are important and can be fostered in school contexts through SEL programs. The present paper analyzed children’s perceptions of using serious games as a complement to the social–emotional activities of the ‘Me and the Us of Emotions’ program. Our quantitative and qualitative findings converge to suggest the usefulness of adding serious games as a complement to a social–-emotional skills-based program, as they contribute to enhancing children’s satisfaction. Given the importance of SEL programs and the benefits they have been found to provide [1,4,7], it seems relevant to continually improve the satisfaction and responsiveness of children toward these initiatives. Our findings add to the previous literature [27] in suggesting that serious games be used as an increasingly more common component of SEL programs, particularly those directed at younger generations. In addition, participant responsiveness and its relationship to the outcomes of social–emotional programs could offer useful information for schools and, therefore, reinforce the importance of implementing social–emotional learning programs within those contexts.

## Figures and Tables

**Table 1 ijerph-19-09613-t001:** Model summary for regression analyses using satisfaction with sessions that did and did not use serious games as independent variables and enjoying activities and interest in subjects as dependent variables, respectively (N = 57).

	R^2^	Adjusted R^2^	B	*t*
Model predicting enjoyment	0.418	0.397		
Satisfaction with sessions that used serious games			0.753	4.54 ***
Satisfaction with sessions that did not use serious games			−0.145	−0.87
Model predicting interest	0.362	0.338		
Satisfaction with sessions that used serious games			0.516	2.96 **
Satisfaction with sessions that did not use serious games			0.105	0.60

Note. ** *p* < 0.01. *** *p* < 0.001.

**Table 2 ijerph-19-09613-t002:** Number of participants and references in each sub-themes and examples (N = 221 participants).

Name	Participants/ References	Examples
Positive aspects		
Learning emotion regulation strategies	59/74	“*I learn the turtle technique. Because I can think, and I think it would be fun for everyone to have somewhere to hide to think*.”
Experiencing positive emotions		
Playing activities	19/24	“*What I liked the most were the activities because we did things together*.”
Serious games	107/126	“*Games because they help to understand how to deal with emotions*.”
Enjoyment	173/234	“*I really enjoyed the activities because they were fun*.”
Shared experience	3/3	“*I enjoyed being together*.”
Gaining emotional knowledge		
Awareness of others’ emotions	6/6	“*I also love talking about emotions, it makes me feel like we all have the same issues*.”
Body awareness	2/2	“*I can learn how to deal with our bodies*.”
Emotional expression	125/153	“*It was interesting to learn more about emotions and also learn new emotions*.”
Negative aspects		
Reduced opportunity for playing serious games	12/13	“*I enjoyed playing it less often*.”
Experiencing difficult emotions	45/45	“*What I liked least about the program was when we were talking about unpleasant emotions because it made me sad*.”
Opportunities for improvement		
Less noise during sessions	11/11	“*My advice is to have less noise during the sessions*.”
Behaving better in the classroom	11/11	“*People should behave better*.”
Addressing more emotions	10/10	“*I would like to learn more and other emotions*.”
More serious games	9/9	“*My advice is to show more games*.”
Increasing session duration	17/18	“*My advice is the sessions last longer*.”

## Data Availability

The data presented in this study are available on request from the corresponding author. The data are not publicly available due to the study being ongoing.

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
