# Peer review of "Children’s Perspectives on Using Serious Games as a Complement to Promoting Their Social–Emotional Skills"

_ijerph, 2022, doi:10.3390/ijerph19159613_

Round 1

Reviewer 1 Report

Reviewer Report

Paper: Children’s perspective on using serious games as complement to promoting their social emotional skills 

The paper investigates the use of serious games may be an appealing and complementary way to motivate curriculum-based social and emotional learning (SEL), through the "Me and We of Emotions" programme designed for children. 

The paper carries out a quantitative and qualitative assessment of the degree of satisfaction of the participants.

Analyses differences in children ́s satisfaction between sessions that used or did not use serious games. Participants were 232 children (122 boys 21 and 110 girls), aged between 8 and 12 years old.

The results showed similar levels of satisfaction between sessions that did and did not use serious games as a complement to the programme. Satisfaction with serious games was significantly and positively associated with fun, easiness, utility to understand the session and to manage emotion. 

MAIN CONCERN:

1.Update the literature and include contemporary studies, especially from the last three or four years

2. Establish the relevance of the study

3. We do not understand if you use a control group, because you do not expose the results.The purpose of having a control is to rule out other factors that may influence the results of an experiment. The interest of using experimental group and control group is to compare the results.

4. Include a conclusions section

MINOR COMMENTS

1. Authors should check the keywords: one of them (responsiveness assessment) is only included in the abstract

Reviewer 2 Report

This is an interesting article that I am sure is an original idea and certainly of interest to content and program developers of educational/pastoral tools.

I personally was more interested in the impact of serious games on the effectiveness of the sessions on the participant's emotional development rather than satisfaction and since you do note the positive impact of sessions including games on this, it might be worth paying some further attention to this in your discussion. It might be worth pointing out that this could be an area for future research. Also as much as your study is well designed and outlined here, it would be good to provide more detail on the purpose of the study and who should benefit from it. Is there an intended audience? Is the audience commercial, educational, academic?

There are a number of typing errors that should be amended at this stage. 

I enjoyed reading the article however, and hope that subject to some amendments based on the above, it will be soon ready for publication.

Reviewer 3 Report

Thanks for providing me the opportunity to review the manuscript. The suggestions are listed in the following aspects:

1)      It is not stated the exact age (year and months) of the children in the abstract.

2)      Need to clearly define “serious games” in the article.

3)      Did parents and teachers sign the consent forms?

4)      Strengthening the discussion part
